# CERTAINTY IN, CERTAINTY OUT
# REVQCS FOR QUANTUM MACHINE LEARNING

## ABSTRACT

The field of Quantum Machine Learning (QML) has emerged recently in the hopes of finding new machine learning protocols or exponential speedups for classical ones. Apart from problems with vanishing gradients and efficient encoding methods, these speedups are hard to find because the sampling nature of quantum computers promotes either simulating computations classically or running them many times on quantum computers in order to use approximate expectation values in gradient calculations. In this paper, we make a case for setting high single-sample accuracy as a primary goal. We discuss the statistical theory which enables highly accurate and precise sample inference, and propose a method of reversed training towards this end. We show the effectiveness of this training method by assessing several effective variational quantum circuits (VQCs), trained in both the standard and reversed directions, on random binary subsets of the MNIST and MNIST Fashion datasets, on which our method provides an increase of $10 - 15\%$ in single-sample inference accuracy.

## 1 INTRODUCTION

QML research has seen a boom in the last decade, primarily motivated by the exponential speedups offered by algorithms such as Shor's factorization and Grover's Search (Schuld & Petruccione, 2021; Nielsen & Chuang, 2010). Many would like to realize this kind of speedup for machine learning, though barren plateaus and the difficulty of retrieving the expectation values needed to update parameters during training have been challenging obstacles to overcome (Tilly et al., 2022). These challenges are further corroborated by the variance in outputs obtained from noisy intermediate-scale quantum (NISQ) devices and by how this noise increases the number of samples needed to approximate expectation values on modern quantum hardware (Schuld & Petruccione, 2021; Tilly et al., 2022).

Two common methods researchers use to temporarily overcome these challenges are high sampling rates and classical simulation, which provide a means to approximate or calculate an expectation value, respectively, and to reduce the effect of noise (Kohda et al., 2022; Farhi et al., 2014; Skolik et al., 2022; Bowles et al., 2023). Neither of these methods offer a speedup, though they do allow researchers to design algorithms which will run on the quantum computers of tomorrow and to discover fundamentals about how quantum computers will learn, as though the NISQ era were behind us. One issue we see in previous work (Hur et al., 2022b; Farhi & Neven, 2018), is that expectation values of qubits are compared with a probability threshold in order to train a regressor to make binary decisions, which could be considered common practice in the world of classical machine learning; however, this practice leads to models which must be run this same way (using samples and simulation) during inference time in order to achieve any level of high accuracy. It remains a major challenge in QML to use the discretization of quantum states as a matter of fact, rather than forcing quantum computers to do the same floating point calculations as classical computers.

We focus on binary decision problems, as these are the most elementary to model (Sivia & Skilling, 2011), and they innately work under the assumption of discreteness. We propose to enhance the assessment of model accuracy by incorporating these discrete outputs, that is, to simply sample a QML model once and compare the output to the target value. However, in spite of work by Lee et al. (2021) towards remedying this, QML models still cannot be back-propagated from samples. Training and inference are generally done the same way, so the inability to back-propagate from samples

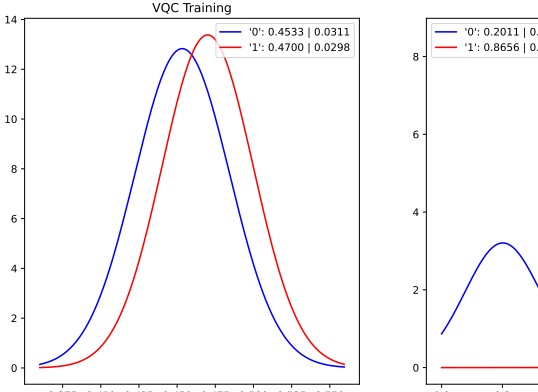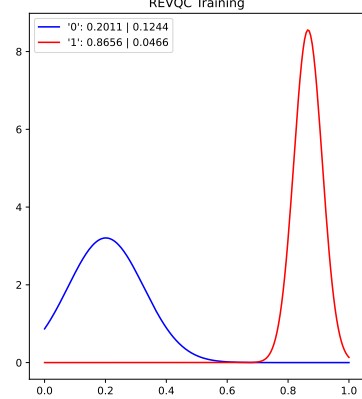

Figure 1: The distribution over expectation values of the predicted targets when trained using *Left:* standard back-propagation. *Right:* REVQC. Both methods use CNN8 from (Hur et al., 2022a)

has, in part, led to researchers abstaining from using a single sample under inference as well, though these processes do not need to be identical. Instead, we propose Reversed Variational Quantum Circuits (REVQCs) as an alternative for training, which are simply the adjoints of VQC circuits, and can be re-reversed during inference. REVQCs still require expectation values for training, with a bit a twist, but achieve low enough epistemic uncertainty to move QML a step toward ignoring expectation values and instead sampling during inference. In order to explain why REVQCs lead to increased sampling accuracy, we also reframe the goal of training a VQC in terms of aleatoric, epistemic, and quantum uncertainty.

In the next section, we will bring up relevant findings from other works. Our major contributions are as follows:

- In Section 3, we introduce the reader to concepts from quantum computation which act as a foundation for our methodologies, present our method for reversed VQC training, and lay out the experiments we ran to test its efficacy versus standard training.

- In Section 4, we present the results of our experiments and make the case for using single-sample accuracy as a standard metric in QML.

- In Section 5, we give our analysis of the effects that aleatoric, epistemic, and quantum uncertainties have on VQC training and on how to minimize them in order to achieve high single-sample accuracy.

## 2    RELATED WORKS

Much of QML research is committed to research-problems which are quantum in nature. Methods such as the Quantum Approximate Optimization Algorithm (Farhi et al., 2014) and Variational Quantum Eigensolver (Tilly et al., 2022) are both parameterized quantum circuits designed to iteratively find the ground state of a Hamiltonian. These methods are very useful for quantum chemistry, condensed matter physics, and graph problems, but they require high-sampling rates for training. Despite that both are well-known uses of parameterized circuits, they do not pertain to discrete outputs, so their domains are outside the scope of this paper.

There is plenty of QML research on classification, much of which has become the primary motivation for the work in this paper. The work on Quantum Convolutional Neural Networks (QCNNs) done by Hur et al. (2022a), especially their overview of VQC ansatz and expressive unitaries, laid the groundwork for the experiments done on REVQCs. Several of the unitary circuits used in their work, and subsequently our paper, were first introduced and more deeply analyzed by Sim et al. (2019) and Wei & Di (2012), and the method of dual-angle embedding we use to encode the inputs

comes from LaRose & Coyle (2020). None of these works analyze the effects these ansatz or unitaries have with respect to reversibility, a term used to describe computational processes which do not destroy information (Nielsen & Chuang, 2010). Quantum computation is described with this term because quantum gates are unitary, even when many are used sequentially (Evangelidis, 2021), meaning their use on a quantum state is equivalent to a linear bijection, so it holds that the unitary circuits from these works are bijective. We leverage this bijectivity to learn the mapping from the inputs to the outputs in reverse and reduce uncertainty.

REVQCs can be seen as quite similar to the adjoint method of back-propagation introduced by Jones & Gacon (2020), which provides a means of calculating highly accurate gradients by applying the derivatives of the adjoints of the parameterized gates from a VQC. Both use the adjoints of VQC unitaries during training, however, this method is only an improvement to the calculation of gradients. To this end, adjoint back-propagation is even compatible with REVQCs, though the adjoints used therein are the original gates used in the standard VQC. Adjoint back-propagation and REVQCs both make use of the reversibility property, but the former relies on the same expectation values needed for any other VQC back-propagation and does not affect the epistemic uncertainty of the trained VQC.

Prior work on uncertainty in machine learning focuses mostly on how to model and reduce epistemic uncertainty. An informative overview of the different kinds of uncertainty as it relates to machine learning can be found in Hüllermeier & Waegeman (2021), as well as a classical method to reduce aleatoric uncertainty. Uncertainty is a primary idea in active learning (Aggarwal et al.; Hoarau et al., 2023), where data samples with more class uncertainty are prioritized for labeling unlabeled training data. Modelling uncertainty is often done with ensemble methods (Beachy et al., 2023; Schreck et al., 2023), Bayesian methods (Hüttel et al., 2023), and confidence indicator methods (Trivedi et al., 2023). Since these methods are not focused on a bijective map between inputs and targets, they are not able to manipulate the uncertainty the way REVQCs can.

## 3 METHODS

### 3.1 BACKGROUND

In order to most easily talk about the quantum aspects of this paper, we will lay out some of the terminology used here.

*Qubits* - Qubits are the quantum equivalent to a bit in classical computing. The state of a qubit is represented as a two dimensional vector in a Hilbert space, with classical states 0 and 1 corresponding to the quantum states $|0\rangle$ and $|1\rangle$, where

$$|0\rangle = \begin{bmatrix} 1 \\ 0 \end{bmatrix}, \text{ and } |1\rangle = \begin{bmatrix} 0 \\ 1 \end{bmatrix}. \tag{1}$$

Unlike classical bits which are binary, the state of a qubit can be any length-1 vector in the two-dimensional complex vector space spanned by $|0\rangle$ and $|1\rangle$.

*Gates* - Quantum gates are the quantum equivalent of classical (reversible) gates. These transform states unitarily (complex angle-preserving), so correspond to complex rotations. Simple examples include the Pauli-X, Pauli-Y, and Pauli-Z gates, written mathematically as $\sigma_1$, $\sigma_2$ and $\sigma_3$, respectively. Pauli-X, Pauli-Y, and Pauli-Z are also names for the cardinal axes within the sphere of all possible states a single qubit can take, the so-called "Bloch sphere". The matrices which represent these operations rotate a qubit $\pi$ radians around the respective axis, and all of them can be written at once as,

$$\sigma_j = \begin{pmatrix} \delta_{j3} & \delta_{j1} - i\,\delta_{j2} \\ \delta_{j1} + i\,\delta_{j2} & -\delta_{j3} \end{pmatrix}. \tag{2}$$

Many other gates exist, including gates to elicit interactions between qubits and parameterized versions of the Pauli Gates which allow rotations of arbitrary degree around their respective axes.

*Circuits* - The term "circuit" typically refers to a more complicated unitary operator built up from a number of quantum gates that are composed sequentially. The term *wire* refers to single qubits as they traverse the different operations within a circuit. The term "model" can often be interchanged with "circuit," though perhaps self-evidently, only when the model can be represented as a circuit.

*Reversibility* - In quantum computing, every operation on a circuit has a conjugate inverse, known as an adjoint, that is easily derived, or in simple cases identical to the gate itself. Since these gates are all rotations, this can be understood as rotating on the same axis by the opposite angle. Much of the work in this paper relies on the fact that by learning the parameters for the adjoint of a unitary (a REVQC, for instance), the parameters of the original unitary (the VQC) are also known.

*Measurements* - To extract information from a quantum circuit, a measurement of the qubits involved must be performed. A measurement has an associated Hermitian operator (real-valued eigenvalues) where the eigenvalues are the possible outcomes, and the squared length of the state projection onto one of the eigenspaces determines the probability of the corresponding outcome. Due primarily to convention, the most common measurement in quantum computing is measurement in the "computational basis", associated with the Hermitian Pauli-Z operator (Schuld & Petruccione, 2021; Nielsen & Chuang, 2010).

A measurement always gives one of the eigenvalues of the Hermitian operator. For a Pauli-Z measurement we obtain one of two discrete outputs, +1 or -1 (mapped to the bit values 0 or 1, respectively). The *expectation value*, or the expected (average) output is then the weighted average of the outcomes. For a Pauli-Z measurement, this would be

$$E(\sigma_3) = (+1)P(+1) + (-1)P(-1) \tag{3}$$

Note that the range of this expression is $[-1, +1]$ because the eigenvalues of the $\sigma_3$ operator are $+1$ and $-1$ rather than the binary 0 and 1. If the two outcomes are equally probable, the expectation value here is =0 rather than =1/2, which we will need to remember later when we set thresholds in the simulation output.

Such an expectation value can be calculated directly, though this is only possible in simulations. In an actual machine, the outputs would be the discrete values $+1$ and $-1$, so to estimate the expectation value when using a quantum computer one would need to count the outcomes and produce a point sample from a series of measurements.

It is important, in the context of this work, to view the expectation value as a measurable quantity output by a quantum model, and not merely a statistic used to describe the quality of said model, without forgetting that it still describes a statistical moment. This ambiguity is a side effect of the collapsing of quantum states, and the disambiguation of these two definitions is part of what makes sample accuracy so important. A model can be deemed accurate if its mean output is higher than a certain threshold, as is the goal of most QML research, but this does not mean that outputs will be precise. A precise model will output correct samples more often than not, which promotes pushing the expected values of a model away from the threshold and also reducing the variance.

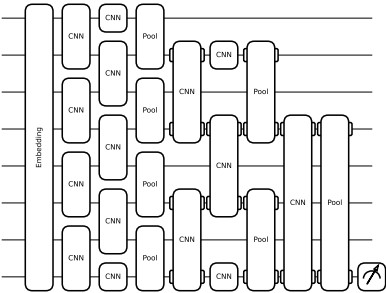

Figure 2: Full circuit used for all tests. Pooling stays the same but CNNs are switched out depending on the selected circuit.

## 3.2 SETUP

For our experiments we leverage the highly-expressive Quantum Convolutional Neural Network (QCNN) model and training structure used by Hur et al. (2022a), though we re-implement it in PyTorch (Paszke et al., 2019) for parallelization and thus faster training. The tools for doing QML,

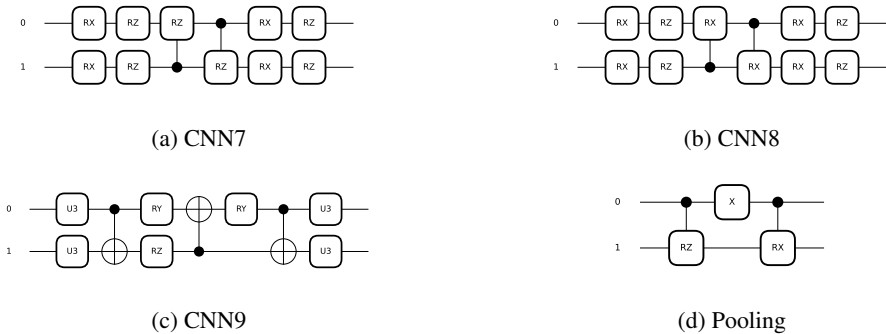

(a) CNN7                                   (b) CNN8

(c) CNN9                               (d) Pooling

Figure 3: Unitary circuits used in the QCNN.

including the ability to create the adjoint circuits used in REVQC, are provided by Pennylane (Bergholm et al., 2022) and function well in PyTorch. The base circuits used in this set of experiments were proposed by Sim et al. (2019) and Wei & Di (2012). The full circuit diagram for the QCNN can be seen in Figure 2, wherein the way the constituent parts described below are put together. In all tests, we use the same QCNN circuit setup and input embedding, MSE-loss (explained below), and simply swap out the CNN unitaries. We will use *VQC* to refer to the QCNN and *REVQC* to refer to the adjoint of the QCNN.

*Input Embedding -*The full VQC circuit is comprised of eight qubits with a single output qubit. The representation of the images inputted to the standard model are created by doing PCA on the image dataset, taking the top 16 values, normalizing these values, and multiplying them by $\pi$ (Hur et al., 2022b). We encode the first eight values as angles around the Pauli-X axis, and the next eight as angles around the Pauli-Y axis. This process is named dense angle-embedding by those who first described it, LaRose & Coyle (2020).

*CNN & Pooling Circuits -* The three different two-qubit CNN unitaries, and the two-qubit pooling unitary we use are displayed in Figure 3. These unitaries had the best performance according to Hur et al. (2022b), with higher integers in the names equating to higher expressibility and entangling ability for the circuits. These two qualities allow these circuits to act much like a classical convolutional kernel, finding patterns in the input data to pass high level information further down the circuit. The pooling circuits function similarly to classical pooling layers as well, combining the information from two wires to one wire for the other to become ignored. As our contribution is our training method and not improved unitaries, we use them as is.

*Loss Calculation -* For both training directions, we make use of MSE-loss, which minimizes the L2-distance. For VQC training, the predictions come from the Pauli-Z expectation on the first wire, converted to a probability using 3, as in Hur et al. (2022b), though they use cross-entropy loss instead. We opted to consistently use MSE-loss instead as this was necessary for REVQC training. In REVQC training, we take a Pauli-Z measurement of each of the eight wires, and do MSE-Loss against the ground-state, meaning every expectation value should be as close to 1 as possible. In cross entropy loss, the total of all predictions should be equal to 1, and only one target value should be 1, which does not suit qubit measurements.

The setup of the VQC used in the QCNN involves starting with a number of qubits equal to a power of two, and applying a series of layers with one set of parameters for each CNN, and one for each pooling. In each layer, the CNN circuit is applied to each neighboring pair of qubits, followed by pooling on every other pair, which reduces the number of wires by half. This means that for our experiments, we have three such layers. In the last layer, a single CNN and pooling layer leave just one wire, which is measured to produce an output. When training the standard QCNN, the full circuit is comprised of the input embedding followed by the VQC. In our method, the reversed circuit is comprised of the target encoding, then the REVQC, and finally the adjoint of the input embedding. Even though the task is to train the REVQC to output the image-values given the targets, during training, the adjoint of the input embedding is used to avoid needing to measure twice for each input, ie. the expectation values over both the Pauli-X axis and the Pauli-Y axis. Both circuits use the standard setup for inference.

### 3.3 EXPERIMENTS

Using both the MNIST Digit Recognition Dataset (Deng, 2012) and the MNIST Fashion Dataset (Xiao et al., 2017), we randomly select five pairs of classes from the ten available (without replacement), then train either the VQC or the REVQC for binary classification over the respective pairs. We compare the accuracy as calculated from the expectation value on the first wire from both training schemas as well as the accuracy as calculated from a single sample for both. For the calculation of the expectation value accuracy, values above 0 are selected as belonging to the first randomly selected class, and values below 0 are selected as belonging to the second randomly selected class. Sampling accuracy is similar, as the model either outputs 1 or $-1$ which correspond to the first or second randomly selected class, respectively. For each of the three best unitaries mentioned previously, and each subset of classes, we repeat this process three times. The full spread of comparisons can be found in the Appendix, and macrostatistics over the subsets can be found in the next section.

## 4 RESULTS

To fully understand the metrics used in the qualitative results, it is important to understand the difference between the two types of possible measurement outputs from a VQC, as explained in section 3.1. In all the tables, we show the results from assessing the fifteen models (three random initializations each for the five random subsets) trained as both VQCs and REVQCs using the three selected QCNN unitaries mentioned in 3.3. In Tables 1 and 3, we show the assessments when a single-sample is output, and in Tables 2 and 4, we show the assessments when an expectation value is output.

The mean and standard deviation are calculated across the average number of correct binary classifications for the fifteen models trained in each category. The accuracies for VQC training are universally lower, for both types of inference and both datasets, quickly demonstrating the strength of REVQCs. We see that, expectedly, the accuracies tend to go up as the expressivity of the unitary used for training goes up. The standard deviation is quite high in most columns due to some class subsets being much harder to tell apart than others, though much lower when sampling the VQC trained models as the expectation values of the states learned by these circuits tend toward 0. This will be discussed slightly in a few paragraphs, and even more in Section 5.

|  | VQC | REVQC |
|---|---|---|
| CNN7 | $50.35\% \pm 3.58$ | $64.73\% \pm 9.14$ |
| CNN8 | $49.86\% \pm 1.77$ | $65.17\% \pm 6.96$ |
| CNN9 | $50.09\% \pm 2.75$ | $\mathbf{66.25\% \pm 3.26}$ |

Table 1: MNIST Sampling Accuracies

|  | VQC | REVQC |
|---|---|---|
| CNN7 | $52.62\% \pm 16.52$ | $79.88\% \pm 14.97$ |
| CNN8 | $52.84\% \pm 13.34$ | $84.61\% \pm 12.31$ |
| CNN9 | $50.82\% \pm 10.00$ | $\mathbf{93.8\% \pm 3.03}$ |

Table 2: MNIST Expectation Value Accuracies

|  | VQC | REVQC |
|---|---|---|
| CNN7 | $48.47\% \pm 2.66$ | $64.08\% \pm 8.32$ |
| CNN8 | $49.17\% \pm 1.64$ | $61.40\% \pm 8.57$ |
| CNN9 | $50.44\% \pm 1.91$ | $\mathbf{69.57\% \pm 6.13}$ |

Table 3: Fashion Sampling Accuracies

|  | VQC | REVQC |
|---|---|---|
| CNN7 | $49.49\% \pm 13.4$ | $77.59\% \pm 12.87$ |
| CNN8 | $44.57\% \pm 12.16$ | $80.43\% \pm 14.44$ |
| CNN9 | $46.13\% \pm 17.41$ | $\mathbf{94.92\% \pm 4.21}$ |

Table 4: Fashion Expectation Value Accuracies

It is interesting to note that with VQC training, the expectation value accuracy is sometimes, counterintuitively, lower than the sampling accuracy. When analyzing the expectation accuracies, what we average are the per-input-sample prediction confidences. The higher standard deviations seen across the expectation category point toward a reason for the discrepancy. A single prediction with an expectation value of .5, when compared with a threshold of 0.0, will always be correct. When this exact quantum state is sampled, there remains a 25% chance the sample will be incorrect.

Furthermore, in the event that the model does a poor job of separating the two classes, their expectation values could become equal. Take the hypothetical extreme case where this happens in a

model such that it mapped every input to the same example expectation value as before. The expectation value accuracy would be $50\%$ as every $|0\rangle$ class would be labeled correctly, and every $|1\rangle$ class incorrectly. The sampling accuracy could then stochastically become slightly higher than this, if the $25\%$ chance of a $|1\rangle$ input outputting the correct value happens more frequently than the equal-probability opposite case. Examples of this can be seen in the APPENDIX.

Poor intra-class separation is one thing that is exposed by sampling stochasticity, but stochasticity issues are most likely to occur when the the predicted expectation for a given input class is close to $0$. In these cases, model confidence, and thus sampling accuracy, are only slightly higher than guess-level, though high enough to still achieve high expectation accuracy, since the cutoff is $0$. Neither poor class-separation nor low-confidence are valuable qualities for a machine learning model to have, thus we argue that single sample accuracy is an important metric to assess when determining the quality of a QML model. Using metrics other than single-sample accuracy to determine the quality of a model hides these problems more than it solves them.

## 5 ANALYSIS

When training a supervised machine learning model, the primary goal is to use a loss function to adjust the model weights such that their predicted class labels are approximately equal to a set of labeled targets. The discrepancy within the approximation stems from well-documented forms of uncertainty known as epistemic (model) uncertainty, and aleatoric (data) uncertainty (Hüllermeier & Waegeman, 2021). While aleatoric uncertainty exists in the data and cannot be reduced, epistemic uncertainty comes from a lack of knowledge, and is thus reducible through training. This means that the best model we can train needs to have minimal epistemic uncertainty in the face of intrinsic aleatoric uncertainty. An accurate model minimizes the epistemic uncertainty well enough to map the aleatoric uncertainty (and by extension variance) in the data to the targets such that the variance over the predicted targets is also minimal.

The nature of modern quantum computers introduces even more uncertainty to QML. Noise on quantum hardware creates aleatoric, and thus, irreducible uncertainty, which is separate from that induced by the data itself. Furthermore, measuring a quantum state is a stochastic sampling event that gives only discrete values. This state-collapse introduces further aleatoric uncertainty if not partially or fully circumvented by multi-shot expectation approximation or simulation, respectively. The former turns the number of shots into a hyper-parameter that scales the expectation value estimation precision proportionally to the run-time (Tilly et al., 2022), whereas the latter removes this uncertainty altogether yet limits the scale of calculations to those which are simulatable. Avoiding this sampling uncertainty by reducing the efficiency of quantum methods is not a reasonable long-term strategy in QML. Reducing the output uncertainty, and thus variance, should be a primary goal. In order to reach this goal, two unique traits of quantum computation need to be leveraged: mixed states and reversibility.

In quantum computing, we can describe a circuit as being in one of two kinds of state: pure or mixed. A pure state can be represented as a vector in the Hilbert space spanned by the qubits of the circuit. A mixed-state is a quantum state which is a stochastic mixture of pure states and can only be represented as a density matrix. When looking at a VQC, the inputs are, in essence, a type of external stochasticity to the circuit, meaning the model can be represented as a mixed-state. This turns the VQC into a positive semi-definite operator acting upon the inputs. Parameterizing this operator will then allow back-propagation to incrementally adjust the relationship between the inputs and targets.

Quantum models are not universal function approximators (Schuld & Petruccione, 2021) and will always be linear on pure states. Because of this, VQCs tend to take input uncertainty and turn it into output uncertainty. This is because of variance contained in the large number of similar pure-states the VQC density matrix is ultimately forced to parameterize all at once. Variance in the inputs leads to variance in the outputs, which in turn leads to varied gradients. Due to the fact that quantum information and operations are all $2\pi$-periodic, high variance in the gradients leads to them cancelling each other out. This in turn pulls the model predictions toward the center point of the output space, as we can see in the left-hand side of Figure 1.

The problems incurred by training a mixed state on highly uncertain inputs can be rectified by making use of the reversibility of quantum computations and the fact that a pure state is a bijec-

tion. In classification problems, the targets are discrete. An ideal training method, devoid of back-propagation restraints, would output predictions with no variance, and could be perfectly visualized as a histogram instead of a distribution. On the contrary, a high degree of variance is accepted, if not expected, in the input space. Thus, rather than training a mixed state to map from high-variance inputs to low-variance outputs, map instead from low-variance inputs to high-variance outputs. The REVQC is still a linear mapping from the input states to the output states, so rather than transferring the input variance to the target space as a standard VQC does, REVQC can force its predictions to have minimal variance as compared to the samples in the input-space. In effect, REVQC learns the exact center point in the sets of Hilbert space vectors associated with each class. These center points and their relationship to more precise inference will be explained in 5.2.

## 5.1 VISUALIZATION OF MODEL ACCURACY

The only measure of uncertainty which is possible when comparing discrete outputs to discrete targets is accuracy. Unfortunately, accuracy is non-differentiable on a per-input basis. This is why classical machine learning models and QML models alike use continuous values for training and threshold the same continuous values for accuracy. One upside to this methodology is being able to build a probability distribution over the soft outputs, rather than just a histogram. This type of output distribution is shown in Figure 1 for a regular VQC and a REVQC trained using the same circuit.

While the basis for REVQC's quantitative improvements lies in its ability to output better single-sample results, a visualization of the expectation values of both VQCs and REVQCs does a good job of showing how much more precise the learned mapping is when using REVQCs. In Figure 1, it can be seen that the chance of making a false prediction is much smaller when using REVQCs, and that the variance over the expected target predictions from REVQC is such that even in the worse case, the model is quite likely to select the right class. This solidifies the notion that during training, certainty in leads to certainty out, thus training a VQC on low variance inputs has a major effect. The variance in expectation values when inferring with the VQC trained in reverse come from aleatoric input uncertainty, that is, the distance from the input vectors to each other. This corresponds to VQCs being equivalent to kernel methods (Schuld & Petruccione, 2021).

## 5.2 VISUALIZATION OF RECEPTIVE FIELDS

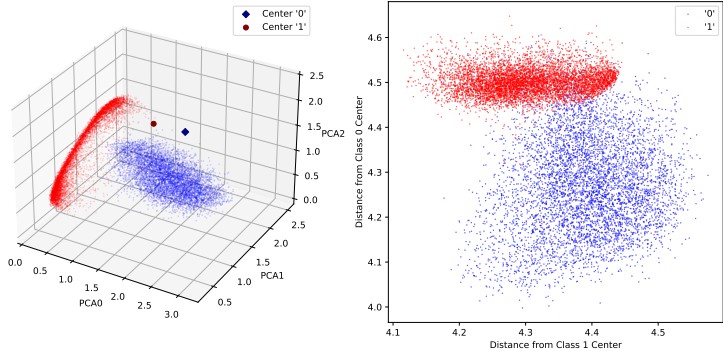

Figure 4: These two figures visualize the receptive field learned by standard VQC training. *Left:* First three PCA dimensions representation of the inputs as well as the center point learned by the VQC. *Right:* Scatter-plot displaying the distances from each center point to each input in the dataset. Values marked as '0' should be closer to only the X-axis, and values marked as '1' should be closer to only the Y-axis.

To visualize the difference in receptive fields between a standard VQC and a REVQC, we have created a series of paired visualizations. Figures 4 and 5 show these paired visualizations for CNN8 from Hur et al. (2022a) trained as a VQC and as a REVQC, respectively. For the visualization on the left of the figures, we have plotted the spread of the inputs with respect to the top three (of sixteen)

PCA dimensions used in the training. The plots also show the center points learned by the model for each of the binary input values. Given that there are only two possible classes, and a quantum model is linear for pure states, this means there is only one possible point that could be generated for each input class, the $|0\rangle$-center and the $|1\rangle$-center. To generate these, we ran both circuits in the reversed direction, once for each target value, without the input embedding, then output the Pauli-X and Pauli-Y expectation values. These values are then mapped to the value range of the input by normalizing them to the range $[0, \pi]$. It can be seen that the model trained using our REVQC method learns more precise center points for separating the classes.

The plots on the right-hand sides of Figures 4 and 5 are an extension of those on the left-hand side showing the angular distance from each input vector to the $|0\rangle$- and $|1\rangle$-center points on the $Y$-axis and $X$-axis, respectively. These graphs were generated by inferring all data samples in the reversed direction with input embedding once for class label in the target encoding. We measure the Pauli-Z expectation values of all eight qubits and take their $\arccos$ to determine the angles from the ground state, then take the magnitude of this vector. In an ideal binary classification circuit, the combination of the input embedding and VQC would always output $|0\rangle$ or $|1\rangle$, depending on the class. In the case of the reversed circuit, where we encode both the inputs and the targets, we would expect to go from $|0\rangle$ or $|1\rangle$ back to the ground-state. Thus this angle represents a classification error of the REVQC. Inputs with lower values on their respective axes are more likely to output the correct label when sampled during VQC inference. The behaviour we expect to see when a VQC is doing a good job of distinguishing the two classes is shown more closely in Figure 5, where the distances to the incorrect label are high, and the distances to the correct label are low.

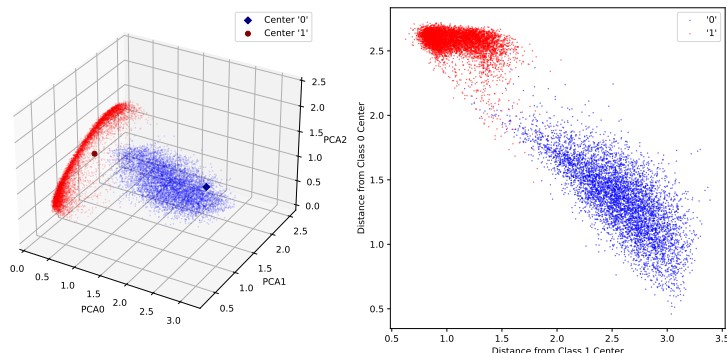

Figure 5: These two figures visualize the receptive field learned by REVQC training. *Left:* First three PCA dimensions representation of the inputs as well as the center points learned by the REVQC. *Right:* Scatter-plot displaying the distances from each center point to each input in the dataset. Values marked as '0' should be closer to only the X-axis, and values marked as '1' should be closer to only the Y-axis.

## 6  CONCLUSION

In this paper, we have discussed sample accuracy as a metric worth being more aware of in quantum computing, and shown that it can be improved if careful consideration is given to the uncertainties intrinsic to the data and hardware. We presented a novel means of reducing epistemic uncertainty by taking the quantum uncertainty out of the target predictions and allowing it to be combined with the aleatoric input certainty. The presented method involves little more than training the parameters of a VQC in reverse by instead using its adjoint. For the case of binary classification, we have demonstrated that this allows the parameters to learn a better receptive field over the input space which also leads to more accurate and precise output predictions under the single-sample assumption.

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

# A   APPENDIX

Tables showing the entire spread of tests over the random binary subsets chosen from MNIST and Fashion MNIST.

| Test | Digits | Sample Acc | ExpVal Acc |
|---|---|---|---|
| CNN7 Standard | 5, 8 | $47.94\% \pm 2.32$ | $43.76\% \pm 6.22$ |
| CNN7 Standard | 0, 3 | $49.51\% \pm 3.09$ | $58.86\% \pm 9.28$ |
| CNN7 Standard | 6, 7 | $50.72\% \pm 0.59$ | $55.58\% \pm 8.39$ |
| CNN7 Standard | 1, 4 | $52.18\% \pm 7.77$ | $42.16\% \pm 31.34$ |
| CNN7 Standard | 9, 2 | $51.38\% \pm 2.01$ | $62.73\% \pm 19.06$ |
| CNN7 Reversed | 5, 3 | $57.15\% \pm 0.70$ | $66.96\% \pm 3.07$ |
| CNN7 Reversed | 6, 0 | $70.14\% \pm 0.89$ | $87.76\% \pm 1.90$ |
| CNN7 Reversed | 1, 4 | $77.78\% \pm 1.92$ | $97.56\% \pm 0.51$ |
| CNN7 Reversed | 7, 9 | $52.34\% \pm 0.33$ | $57.92\% \pm 1.35$ |
| CNN7 Reversed | 8, 2 | $66.25\% \pm 1.69$ | $89.21\% \pm 0.11$ |
| CNN8 Standard | 2, 0 | $50.65\% \pm 1.73$ | $50.75\% \pm 8.82$ |
| CNN8 Standard | 5, 4 | $49.08\% \pm 1.00$ | $57.72\% \pm 18.37$ |
| CNN8 Standard | 7, 8 | $49.62\% \pm 1.49$ | $51.07\% \pm 16.65$ |
| CNN8 Standard | 6, 1 | $50.06\% \pm 1.53$ | $52.93\% \pm 18.90$ |
| CNN8 Standard | 3, 9 | $49.86\% \pm 3.59$ | $51.73\% \pm 15.37$ |
| CNN8 Reversed | 7, 6 | $66.16\% \pm 12.46$ | $82.05\% \pm 25.52$ |
| CNN8 Reversed | 3, 2 | $65.45\% \pm 2.31$ | $85.99\% \pm 2.64$ |
| CNN8 Reversed | 1, 4 | $73.68\% \pm 3.35$ | $98.04\% \pm 0.44$ |
| CNN8 Reversed | 0, 5 | $61.30\% \pm 0.36$ | $79.41\% \pm 3.84$ |
| CNN8 Reversed | 8, 9 | $59.25\% \pm 2.43$ | $77.57\% \pm 8.10$ |
| CNN9 Standard | 1, 5 | $48.57\% \pm 0.71$ | $55.93\% \pm 5.41$ |
| CNN9 Standard | 7, 8 | $50.35\% \pm 1.71$ | $53.60\% \pm 9.55$ |
| CNN9 Standard | 4, 6 | $48.65\% \pm 3.26$ | $41.46\% \pm 10.75$ |
| CNN9 Standard | 3, 9 | $49.58\% \pm 1.10$ | $42.79\% \pm 8.51$ |
| CNN9 Standard | 0, 2 | $53.29\% \pm 4.35$ | $60.31\% \pm 5.31$ |
| CNN9 Reversed | 1, 3 | $71.35\% \pm 3.87$ | $96.40\% \pm 1.26$ |
| CNN9 Reversed | 9, 2 | $65.03\% \pm 2.48$ | $93.94\% \pm 1.28$ |
| CNN9 Reversed | 0, 6 | $65.70\% \pm 1.13$ | $94.80\% \pm 1.06$ |
| CNN9 Reversed | 4, 8 | $63.75\% \pm 1.30$ | $92.28\% \pm 2.86$ |
| CNN9 Reversed | 7, 5 | $65.40\% \pm 1.92$ | $91.58\% \pm 5.81$ |

Table 5: Mnist Full Spread

| Test | Digits | Sample Acc | ExpVal Acc |
|---|---|---|---|
| CNN7 Standard | 9, 5 | $48.17\% \pm 1.27$ | $47.24\% \pm 3.22$ |
| CNN7 Standard | 4, 7 | $47.67\% \pm 4.49$ | $53.70\% \pm 31.64$ |
| CNN7 Standard | 1, 3 | $48.40\% \pm 3.07$ | $52.21\% \pm 10.79$ |
| CNN7 Standard | 0, 6 | $50.04\% \pm 1.17$ | $52.87\% \pm 1.20$ |
| CNN7 Standard | 2, 8 | $48.09\% \pm 3.94$ | $41.41\% \pm 7.59$ |
| CNN7 Reversed | 2, 0 | $53.56\% \pm 1.03$ | $59.50\% \pm 4.49$ |
| CNN7 Reversed | 6, 3 | $58.42\% \pm 0.83$ | $66.66\% \pm 1.23$ |
| CNN7 Reversed | 9, 8 | $66.45\% \pm 3.79$ | $86.69\% \pm 6.28$ |
| CNN7 Reversed | 1, 4 | $77.76\% \pm 0.35$ | $92.87\% \pm 1.09$ |
| CNN7 Reversed | 7, 5 | $64.21\% \pm 0.58$ | $82.22\% \pm 1.73$ |
| CNN8 Standard | 5, 6 | $48.34\% \pm 1.42$ | $32.54\% \pm 5.59$ |
| CNN8 Standard | 8, 0 | $50.11\% \pm 1.85$ | $41.38\% \pm 15.23$ |
| CNN8 Standard | 3, 7 | $49.38\% \pm 2.00$ | $51.43\% \pm 4.21$ |
| CNN8 Standard | 1, 2 | $50.00\% \pm 1.53$ | $54.26\% \pm 16.35$ |
| CNN8 Standard | 4, 9 | $48.05\% \pm 1.74$ | $43.22\% \pm 10.69$ |
| CNN8 Reversed | 1, 3 | $54.74\% \pm 2.41$ | $75.32\% \pm 15.69$ |
| CNN8 Reversed | 9, 2 | $73.55\% \pm 0.36$ | $94.54\% \pm 3.20$ |
| CNN8 Reversed | 0, 8 | $68.67\% \pm 1.76$ | $91.72\% \pm 1.94$ |
| CNN8 Reversed | 6, 4 | $51.12\% \pm 1.64$ | $58.89\% \pm 7.13$ |
| CNN8 Reversed | 5, 7 | $58.90\% \pm 2.04$ | $81.69\% \pm 4.92$ |
| CNN9 Standard | 7, 8 | $49.71\% \pm 0.63$ | $43.08\% \pm 6.38$ |
| CNN9 Standard | 5, 6 | $50.90\% \pm 1.78$ | $51.90\% \pm 6.62$ |
| CNN9 Standard | 0, 2 | $51.21\% \pm 0.93$ | $57.12\% \pm 12.49$ |
| CNN9 Standard | 1, 4 | $48.62\% \pm 2.85$ | $32.68\% \pm 34.80$ |
| CNN9 Standard | 3, 9 | $51.78\% \pm 2.28$ | $45.88\% \pm 17.45$ |
| CNN9 Reversed | 4, 0 | $61.79\% \pm 0.51$ | $90.47\% \pm 2.58$ |
| CNN9 Reversed | 6, 9 | $70.90\% \pm 1.18$ | $98.52\% \pm 1.13$ |
| CNN9 Reversed | 2, 1 | $72.86\% \pm 1.07$ | $95.89\% \pm 1.17$ |
| CNN9 Reversed | 5, 8 | $64.79\% \pm 2.48$ | $90.12\% \pm 2.37$ |
| CNN9 Reversed | 3, 7 | $77.52\% \pm 5.83$ | $99.58\% \pm 0.34$ |

Table 6: Fashion Full Spread

