# OpenReview forum: "Certainty In, Certainty Out: REVQCs for Quantum Machine Learning"
_ICLR.cc/2024/Conference — ICLR 2024 Conference Withdrawn Submission_

### Official Review · Reviewer_eDnW · 2023-10-20

**Soundness:** 1 poor
**Presentation:** 1 poor
**Contribution:** 2 fair
**Rating:** 3
**Confidence:** 5

**Summary:**

The paper addresses the fundamental issue of statistical uncertainty in QML. It tries to solve this problem by training algorithms that achieve high single sample accuracy. The paper claims to have an algorithm that can optimize for this goal. Some numerical results are presented that show improvements in  dimension-reduced  datasets.

**Strengths:**

Tries to address a fundamental issue in QML. Statistical uncertainty due to shot noise is fundamental in all QML methods. Overcoming this barrier is necessary to make QML scalable in the future.

**Weaknesses:**

1. What is the exact definition REVQC? How exactly is the model trained so as to reduce single sample uncertainty? The authors claim to define this in Section 3. But Section 3 does not contain anything concrete on what this new training method is.

2. The authors repeat the fact that quantum circuits are unitary and hence reversible and this somehow helps in reducing the uncertanity in the output. This is connection is not well explained in the text. Overall, the paper needs substantial rewriting with a lot more mathematical detail.

3. The method is tested on datasets that are projected onto a smaller space using PCA. I understand that this is necessary  for  QML as we don't have actual quantum systems. But results on such small datasets is not enough justification for the claim that these methods are somehow interesting. This makes these types of works very unsuitable for a venue like ICLR. A specialized venue for QML would be more suitable. This work would be suitable for ICLR  If the authors can give some stronger theoretical results regarding their method.

While there might be some interesting in this work, overall the paper does not explain the main contribution well. A significant rewrite is required before peer-review.

**Questions:**

1. Can you show how REVQC is different from standard training of parametrized circuits?

2. Can you clarify how the unitarity of these circuits are useful in reducing uncertainty in the output?

3. In the caption of  Figure 1. How does one train a VQC using back propagation?  The authors rightly claim in the beginning of the paper that this is not possible.

---

### Official Review · Reviewer_Udj4 · 2023-10-31

**Soundness:** 1 poor
**Presentation:** 2 fair
**Contribution:** 1 poor
**Rating:** 3
**Confidence:** 4

**Summary:**

Quantum machine learning models typically identify expectation values of observables as predictions. This increases uncertainty and requires many shots to estimate the prediction accurately. In this paper, the author propose a new method to train a variational quantum circuit: by fixing the labels as inputs of the reverse circuits and train the model to reconstruct the inputs. Experiments are proposed to validate the idea that this procedure leads to single sample accuracy higher than expectation value accuracy.

**Strengths:**

- Identify challenges of VQC training and discuss uncertainty in QML
- Analysis trying to explain why REVQC is better than std VQC

**Weaknesses:**

- Choice of architecture: a QCNN model from [Hur et al] is used for benchmarking VQC. The model is deemed highly expressive, however in table 1-4 this leads to accuracy of about 50% for a binary classification problem (equal to random guessing). In contrast, simple ansatz like [https://www.tensorflow.org/quantum/tutorials/mnist] achieve 85% test set accuracy for a similar task. Also, I do not understand why a CNN structure is used, since the data is produced via PCA I would not expect the label to be invariant under shifts of the inputs.
- Choice of benchmark: the experimental setting of using 8 qubits and MNIST and Fashion MNIST makes it difficult to learn general lessons about the behaviour of ML models. I think that more experiments should be done to assess the soundness of the method.

Minor:
- Section 2: "they do not pertain to discrete outputs". I find this confusing since QAOA aims to converge the quantum state to the bit string that solve the classical optimisation problem.
- Section 3: "(complex angle-preserving)" should be "(norm-preserving)"?
- Section 5: "the model can be represented as a mixed-state". I find this and the discussion around it confusing since it's an operator not a state. Did you mean a completely positive map?
- Section 5: "Quantum models are not universal function approximators". I think that this depends on what space of functions one looks at. In the space of Boolean functions they are - since they generalize reversible computation which is universal. So a clarification might be helpful.
- Duplicate [Hur et al] reference

**Questions:**

- What happens if you use a different architecture for the VQC that achieves higher accuracy than 50%?

---

### Official Review · Reviewer_2wYu · 2023-10-31

**Soundness:** 2 fair
**Presentation:** 3 good
**Contribution:** 2 fair
**Rating:** 3
**Confidence:** 5

**Summary:**

In this work, the authors put forth effective training methods to improve the optimization performance of variational quantum circuit-based machine learning tasks. In particular, the proposed approach demonstrates an outstanding performance in single-sample inference tasks.

**Strengths:**

1. The idea of improving the VQC training efficiency, especially on NISQ devices, is interesting.

2. The authors conduct comprehensive experiments to demonstrate the effectiveness of the proposed methods.

**Weaknesses:**

There are so many weaknesses in this paper, hinging it from being accepted.

1. Some technical claims are not correct.

    (a)  In the Introduction part, the sentence "QML models still cannot be back-propagated from samples" is not correctly claimed. The parameters of QML models like quantum neural networks can be well-adjusted using a back-propagation approach when simulating on a classical CPU/GPU, and a parameter-shift rule can be employed to estimate the gradients for back-propagation.

   (b) In Backgrounds of Methods, the description of "Quantum gates are the quantum equivalent of classical (reversible) gates" is incorrect. The quantum gates can be described as unitary matrices and can be classically simulated on classical computers, but they can be taken as some equivalent classical gates like "And" & "Or" gates.

2. As shown in Figure 2, it is very confusing to use CNN representing a quantum gate. CNN denotes a classical convolutional neural network, but the authors intend to showcase a quantum convolutional neural network.

3. The use of multi-quit CNN can make it difficult to be well-trained on NISQ devices. In particular, they have to suffer from very serious Barren plateau problems. Besides, since multi-qubit can be decomposed as the combination of single-qubit gates, the circuit diagram in Figure can be further optimized.

4. The MSE loss function is not optimal for VQC-based regression, as the MAE loss is a better choice than the MAE. The authors can refer to the theoretical work as:

Ref. Qi, J., Yang, C.H.H., Chen, P.Y. and Hsieh, M.H., 2023. Theoretical error performance analysis for variational quantum circuit based functional regression. npj Quantum Information, 9(1), p.4.

5. In the experimental part, the baseline results of VQC on the MNIST datasets are extremely low, which is not correct. Many works of VQC for classification demonstrate that a VQC can attain much higher accuracy.

Ref. Chen, S.Y.C., Huang, C.M., Hsing, C.W. and Kao, Y.J., 2021. An end-to-end trainable hybrid classical-quantum classifier. Machine Learning: Science and Technology, 2(4), p.045021.

6. The paper devotes much content to the background introduction of quantum technologies, but it does not highlight the main contribution of the proposed methods.

**Questions:**

1. Why are the baseline results of VQC so low?

2. Is a non-linear activation method used for the VQC? If not, how does  the VQC approximate a complicated target function when conducting regression?

---

### Author Response · Authors · 2023-11-20

Dear ICLR Reviewers,

After receiving your comments, particularly the ones asking why the baseline method was so weak, we decided to run further tests to elucidate the major difference between the original and the PyTorch experiments and discovered an unusual software interaction that we need further experiments to mathematically explain.

We used our PyTorch implementation to sample the dataset without replacement and run multiple epochs, which was prohibitively slow in the baseline CPU method. After running tests more similar to our own in the baseline code, we discovered that the difference in results between the original CPU code and the GPU code is significant. This is likely caused both by differences in complex-gradient calculation between the software packages and these small variations in the number of training samples seen during training.

Due to this major, albeit unusual, discrepancy, we have determined it best to withdraw the paper and further study this newfound phenomenon.

We thank you for taking the time to read through our paper and provide feedback, and will use it to produce an even better work at a later date.

Best,
Author Team